# Functional characterization and structural bases of two class I diterpene synthases in pimarane-type diterpene biosynthesis

Baiying Xing[1,3], Jiahui Yu[1,3], Changbiao Chi[1], Xueyang Ma[1], Qingxia Xu[1], Annan Li[1], Yuanjie Ge[1], Zhengdong Wang[1], Tan Liu[1], Hongli Jia[1], Fuling Yin[1], Juan Guo [2], Luqi Huang [2], Donghui Yang[1✉] & Ming Ma [1✉]

Pimarane-type diterpenoids are widely distributed in all domains of life, but no structures or catalytic mechanisms of pimarane-type diterpene synthases (DTSs) have been characterized. Here, we report that two class I DTSs, Sat1646 and Stt4548, each accept copalyl diphosphate (CPP) as the substrate to produce isopimara-8,15-diene (**1**). Sat1646 can also accept *syn*-CPP and produce *syn*-isopimaradiene/pimaradiene analogues (**2–7**), among which **2** possesses a previously unreported "6/6/7" ring skeleton. We solve the crystal structures of Sat1646, Sat1646 complexed with magnesium ions, and Stt4548, thereby revealing the active sites of these pimarane-type DTSs. Substrate modeling and subsequent site-directed mutagenesis experiments demonstrate different structural bases of Sat1646 and Stt4548 for **1** production. Comparisons with previously reported DTSs reveal their distinct carbocation intermediate stabilization mechanisms, which control the conversion of a single substrate CPP into structurally diverse diterpene products. These results illustrate the structural bases for enzymatic catalyses of pimarane-type DTSs, potentially facilitating future DTS engineering and combinatorial biosynthesis.

[1] State Key Laboratory of Natural and Biomimetic Drugs, School of Pharmaceutical Sciences, Peking University, 38 Xueyuan Road, Haidian District, Beijing 100191, China. [2] State Key Laboratory of Dao-di Herbs, National Resource Center for Chinese Materia Medica, China Academy of Chinese Medical Sciences, Beijing 100700, China. [3] These authors contributed equally: Baiying Xing, Jiahui Yu. ✉email: ydhui@bjmu.edu.cn; mma@bjmu.edu.cn

Diterpenoids are a large family of natural products that are extremely heterogeneous in both their chemical structures and biological activities[1,2]. The diverse 20 carbon chemical skeletons of diterpenoids are generated via the catalyses of diterpene synthases (DTSs). There are class I DTS enzymes, which initiate the reactions via heterolytic cleavage of the hydrocarbon-pyrophosphate bond, as well as class II DTSs, which initiate the reactions via protonation of a double bond or an epoxide ring[3,4]. The various catalyses of class I and II DTSs contribute to the structural diversity of diterpenoids, extending for example from the linear phytane through 1, 2, 3-ringed systems up to macrocyclic members. During the diterpenoid family, the pimarane-type diterpenoids are tricyclic members that are widely taxonomically distributed[5,6], with examples including orthosiphol A from the plant *Orthosiphon stamineus* that exerts anti-inflammatory activity[7], deoxyparguerol from the sea hare *Aplysia dactylomela* that is cytotoxic against the P388 leukemia cell line[8], and eutypenoid B from the fungus *Eutypella* sp. D-1 that functions in immunosuppression[9]. Representative pimarane-type diterpenoids with their diverse producing-origins and bioactivities are summarized in Fig. S1. In contrast to the very large number of diterpenoids (~18,000) that have been chemically characterized to date[10], much less is known about the catalytic mechanisms through which DTSs generate this vast structural diversity: only 11 DTSs structures have been solved (Fig. S2), and no structures or catalytic mechanisms have been demonstrated for pimarane-type DTSs.

Herein, we report the functional and structural characterization of two pimarane-type DTSs, Sat1646 and Stt4548, both of which catalyze the biosynthesis of isopimara-8,15-diene (**1**, a known compound) using *normal*-copalyl diphosphate (*normal*-CPP, designated CPP in descriptions below) as the substrate. The two DTSs Sat1646 and Stt4548 were discovered from marine *Salinispora* sp. PKU-MA00418 and soil *Streptomyces* sp. PKU-TA00600, respectively, after screening our in-house bacteria library using a PCR-mediated genome mining method. We used engineered *E. coli* heterologous expression systems for enzymatic assays. Briefly, whereas Stt4548 from *Streptomyces* sp. PKU-TA00600 displays strict substrate selectivity for CPP and produces **1** as its sole product, Sat1646 from *Salinispora* sp. PKU-MA00418 can accept *syn*-CPP, besides CPP, as a substrate to produce the *syn*-isopimaradiene/pimaradiene analogues **2–7** (compound **2** is a new compound and compounds **3–7** are known). We solved crystal structures of Sat1646, Sat1646 complexed with magnesium ions, and Stt4548, thereby revealing key residues defining their active sites. Modeling and structure-based site-directed mutagenesis experiments confirmed the distinct structural bases for Sat1646 and Stt4548-mediated catalysis. Finally, comparisons of Sat1646/Stt4548 with other structurally characterized DTSs showed that stabilization of distinct carbocation intermediates underlies the conversion of a single substrate (CPP) into diverse diterpene products (isopimaradiene/pimaradiene, abietadiene or biformene). Our study demonstrates the catalytic mechanism for pimarane-type DTSs and sets the stage for DTS engineering and combinatorial biosynthesis to obtain highly diverse pimarane-type diterpenoids.

## Results

**Genome mining reveals two diterpenoid biosynthetic gene clusters**. As a part of our long-term research of natural product discovery and biosynthesis, we have constructed an in-house bacteria library of marine and terrestrial origins. We previously reported a genome mining method based on probe-hybridization to investigate the biosynthetic potential of actinomycete strains,

leading to the discovery of three diterpenoids from one *Streptomyces* strain[11]. Another strain prioritization method based on high-throughput real-time PCR was reported, leading to the discovery of six platensimycin (PTM) and platencin (PTN) producers among 1911 actinomycete strains[12]. Inspired by these successes in genome mining, we used a similar PCR screening method to investigate diterpenoids and/or their biosynthetic gene clusters from our in-house bacteria library. Given the large amount of marine-derived bacteria in our library, we redesigned the degenerate primers based on the conserved sequences of genes encoding five *ent*-CPP synthases—Sko3988_Orf2 (accession number BAD86797), Swt1.2 (accession number AEV45183), Swt2.2 (accession number AEW22921), PtmT2 (accession number ACO31276), and PtnT2 (accession number ADD83015) from *Streptomyces* strains[12], and one CPP synthase SaCPS from a *Salinispora* strain (Fig. S3)[13], which is the only class II DTS reported from marine bacterial origin.

Using genomic DNAs as the templates, this PCR screening identified two positive hits from 500 marine bacteria and 650 soil bacteria: one from marine *Salinispora* sp. PKU-MA00418 and the other from soil *Streptomyces* sp. PKU-TA00600 (Fig. S4). We sequenced the genomes of the two strains and bioinformatics analyses by antiSMASH and BLAST identified one candidate diterpenoid biosynthetic gene cluster in each strain (Fig. 1a). The *sat* gene cluster from *Salinispora* sp. PKU-MA00418 is almost identical to the *terp1* cluster from *Salinispora arenicola* CNS-205 (Fig. S5), which has been reported to encode a class I DTS (SaDTS), a class II DTS (SaCPS), and a P450 (CYP1051A1) that together produce isopimara-8,15-dien-19-ol[13]. The *stt* gene cluster from *Streptomyces* sp. PKU-TA00600 encodes a class I DTS (Stt4548), a class II DTS (Stt4542), and a GGPP synthase (Stt4543). Only two hits (including the *sat* cluster with high homology to *terp1*) were discovered from this PCR screening, suggesting the target genes screened from our specific degenerate primers may not be widespread among the 1150 bacteria. The *sat* and *stt* gene clusters afford us an opportunity to investigate the functions of two actinomycetes-originated DTSs, which is notable because much less diterpenoids and/or their DTSs have been discovered from actinomycetes than those from plants and fungi.

**Functional characterization of the DTSs Sat1646 and Stt4548**. Given that we detected no diterpenoids in the fermentations of *Salinispora* sp. PKU-MA00418 or *Streptomyces* sp. PKU-TA00600, we expressed the DTS-encoded genes in *E. coli* heterologous systems. The class I DTS Sat1646 and class II DTS Sat1645 show high sequence identities to SaDTS and SaCPS from *Salinispora arenicola* CNS-205, respectively, differing in only a few residues (Fig. S5). SaCPS was previously shown to catalyze GGPP to produce CPP, which can subsequently be used as a substrate for SaDTS to produce isopimara-8,15-diene (**1**, Fig. 1c)[13].

We generated an engineered *E. coli* strain expressing multiple MEP pathway genes to ensure an adequate supply of GGPP-synthesis precursors (Supplementary Information, Fig. S6)[14], and used this strain to explore the potential enzymatic functions of Sat1645 and Sat1646. We found that recombinant Sat1645 was insoluble in our *E. coli* expression system, so we expressed a previously characterized CPP synthase-encoding *smCPS* from *Salvia miltiorrhiza*[15], as a replacement to supply CPP for exploring Sat1646's function. GC-MS analysis of extracts from the engineered strain showed production of isopimara-8,15-dien (**1**) (Fig. 1b), and this structural assignment was supported by NMR analyses after large-scale fermentation and chromatographic purification (Supplementary Information). Thus, Sat1646 can catalyze production of the same diterpene product as SaDTS.

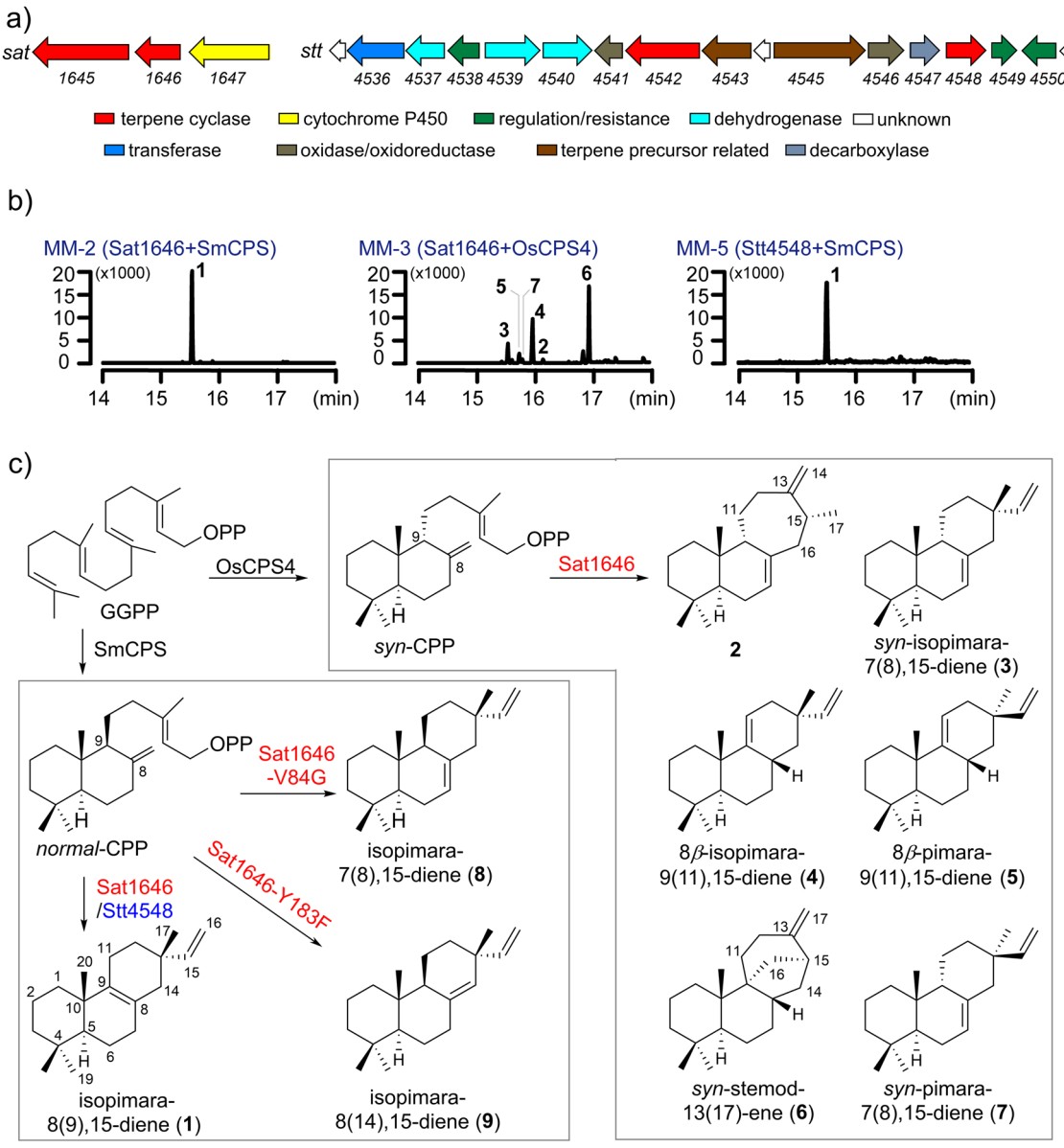

**Fig. 1 The two diterpenoid biosynthetic gene clusters (*sat* and *stt*), the GC-MS analysis of different engineered *E. coli* strains, and the conversion from *normal*-CPP (designated CPP in this study) and *syn*-CPP to diterpenes 1–9 catalyzed by Sat1646, Sat1646 mutants or Stt4548. a** The *sat* and *stt* biosynthetic gene clusters. **b** The GC–MS analysis (extracted ion chromatogram at *m/z* 272) of different engineered *E. coli* strains. The Y-axes show the ion abundances. **c** The generation of **1–9** by Sat1646, Sat1646 mutants or Stt4548.

SaDTS was previously demonstrated to use the substrate *syn*-CPP (provided by a class II DTS from *Oryza sativa*, OsCPS4) to generate isopimarane diterpenes[16]. There are fewer reports of enzymes producing *syn*-pimarane/isopimarane diterpenes as compared to studies of pimarane/isopimarane and *ent*-pimarane/isopimarane diterpene biosynthetic enzymes. To investigate the full *syn*-pimarane/isopimarane diterpene-producing potential of Sat1646 with *syn*-CPP as the substrate, we expressed *osCPS4* in place of *smCPS* with *sat1646* and GC–MS analysis of extracts from this engineered strain revealed multiple candidate diterpene products (Fig. 1b). Large-scale fermentation of this strain and repeated chromatographic purifications enabled structural characterization of *syn*-isopimaradiene/pimaradiene analogues **2–7** (compound **2** is a new compound and compounds **3–7** are known) (Fig. 1c; detailed elucidations in the Supplementary Information). The structural elucidation of **2–7** was benefitted by the structure of *syn*-CPP, whose absolute configurations have

already been set from the catalysis of the class II DTS OsCPS4. Ultimately, crystal structures of **3–6** were solved by X-ray diffraction using Cu Kα radiation (Fig. 2), thereby unambiguously clarifying their absolute configurations. Compound **2** is a tricyclic diterpene with a novel skeleton featuring a "6/6/7" ring system, highlighting Sat1646's capacity to generate previously unknown diterpenes. Compound **3** differs from **4** by its distinct double bond positions, while **3/7** and **4/5** each differ with respect to their C-13 configurations, results suggesting that Sat1646 from the marine strain *Salinispora* sp. PKU-MA00418 can generate a diversity of carbocation intermediates during reactions with *syn*-CPP.

We used a highly similar approach for our functional characterization of the Stt4542 and Stt4548 enzymes from the soil strain *Streptomyces* sp. PKU-TA00600. Again after finding that recombinant Stt4542 was insoluble, we co-expressed *smCPS* with *stt4548* in our engineered *E. coli* strain, which produced

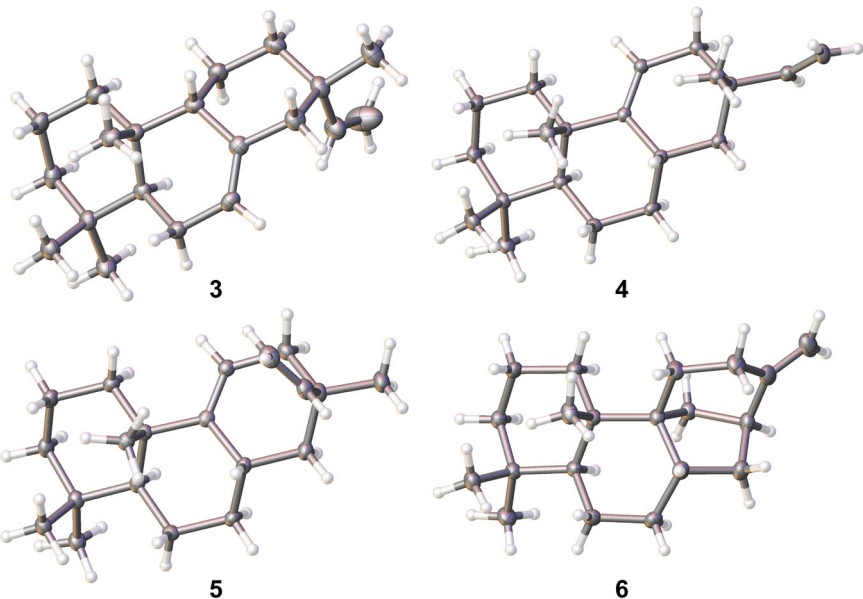

**Fig. 2 The crystal structures of compounds 3–6.** The CIF files of these structures have been submitted as Supplementary Data 4–7.

isopimara-8,15-dien (**1**) as the sole detected product (Fig. 1b). Notably, we found that co-expression of stt4548 with diverse class II DTS-encoding genes including *smCPS* (encoding CPP synthase), *osCPS4* (encoding *syn*-CPP synthase), *ptmT2* (encoding *ent*-CPP synthase), *haur_2145* (encoding KPP synthase), *kgTPS* (encoding *syn*-KPP synthase), and *mtHPS* (encoding TPP synthase) did not support any diterpene production, suggesting that the isopimara-8,15-dien synthase Stt4548 has strict substrate selectivity for CPP.

**Crystal structures of Sat1646 and Stt4548.** As mentioned above, no structures and catalytic mechanisms of DTSs for the bio-synthesis of pimarane-type diterpenoids have been reported. The DTS enzymes AgAS and BjKS are known to produce abietadienes[17] and *ent*-kaurene[18] as major products, respectively, while their mutant variants can produce isopimaradiene[19] and *ent*-pimaradienes[20]. Thus, and recalling our results indicating that Sat1646 and Stt4548 produce only one compound (**1**) when supplied with CPP as the substrate, we were motivated to solving crystal structures for Sat1646 and Stt4548 to support comparative characterization of their catalytic mechanisms. Pursuing this, we conducted crystallization screening for Sat1646 and Stt4548 and successfully obtained high-quality crystals for both proteins. The crystal structures of Sat1646 complexed with magnesium ions (Sat1646-Mg$^{2+}$, 1.94 Å) and apo-Stt4548 (1.57 Å) were solved by Se-SAD (single-wavelength anomalous diffraction) methods, and the crystal structure of apo-Sat1646 (2.50 Å) was solved by molecular replacement using Sat1646-Mg$^{2+}$ structure as the searching model.

The overall structure of Sat1646-Mg$^{2+}$ adopts a classical α-helical bundle fold of class I terpene synthases, with ten core α helices (A–J) and two short α helices (α1 and α2) (Fig. 3a). There is one molecule in one asymmetric unit, and the electron density map was well-defined for residues 2–291 (note that residues 1–3 were encoded by the plasmid sequence and residues 4–299 were encoded by the *sat1646* sequence). Interestingly, Sat1646-Mg$^{2+}$ exhibits a unique crystallographic packing style: because the N-terminal helix A of a first molecule interacts with helices B and J from an adjacent second molecule, the N-terminal loop of the first molecule is actually inserted into the active site of a third molecule (Fig. S9A). Moreover, the

Asn2, Val5, and Thr7 residues of the N-terminal loop of the first molecule form hydrogen bonds with Arg230 at the top of the active site of the third molecule, and the side chain of Met4 in the N-terminal loop of the first molecule forms hydrophobic interactions with residues in the deep active site of the third molecule (Fig. S9B). It seems plausible that these interactions may have promoted the successful crystallization of Sat1646.

We observed two canonical DDXXD and (N,D)DXX(S,T) XXX(E,D) motifs of class I terpene synthase in Sat1646-Mg$^{2+}$: EDWQVD$_{87–92}$ from helix C and NDLASYERD$_{223–231}$ from helix H. Three magnesium ions are present in the active site, including Mg$^{2+}_A$ and Mg$^{2+}_C$ binding to Asp88 and Asp92 from the EDWQVD$_{87–92}$ motif, as well as Mg$^{2+}_B$ binding to Asn223, Asp224, Ser227, Arg230, and Asp231 from the NDLASYERD$_{223–231}$ motif (Fig. S10A). The N-terminal loop from adjacent molecule occupies the space under the three Mg$^{2+}$ ions, with Asn2 close to Mg$^{2+}_A$ and Mg$^{2+}_C$ (Fig. S9C). The crystals of Sat1646-Mg$^{2+}$ were prepared by co-crystallizing Sat1646 with magnesium ions and diphosphates, but no diphosphates were defined due to the presence of Asn2 from adjacent molecule.

The overall structure of apo-Sat1646 possesses similar crystallographic packing style and secondary structure organization as those of Sat1646-Mg$^{2+}$ structure, with ten core α helices (A–J) and one short α helix (α1) (Fig. 3b). The two structures show high similarity to each other, with a root-mean-square deviation (RMSD) of 0.43 Å for 289 Cα atoms (Fig. S10B). However, we observed notable distinctions in the side chain-conformations of residues positioned around the entrance of the active site and the Mg$^{2+}$-binding motifs (Fig. S10A). Further, there was a lack of electron density for this region in the apo-Sat1646 structure that prevented confident building of the Arg169 and Arg233 residues. Finally, the B factor values of residues around the active site entrance in apo-Sat1646, including the two Mg$^{2+}$-binding EDWQVD$_{87–92}$ and NDLASYERD$_{223–231}$ motifs, are much larger than those in the Sat1646-Mg$^{2+}$ structure (Fig. S10C), likely owing to the absence of Mg$^{2+}$ ions in apo-Sat1646.

The overall structure of apo-Stt4548 also adopts the classical α-helical bundle fold with ten core α helices (A–J) and one short α helix (α1) (Fig. 3c). There is one molecule in one asymmetric unit, and the electron density map was well-defined for residues 30–244 and 251–305 (residues 1–21 were encoded by plasmid

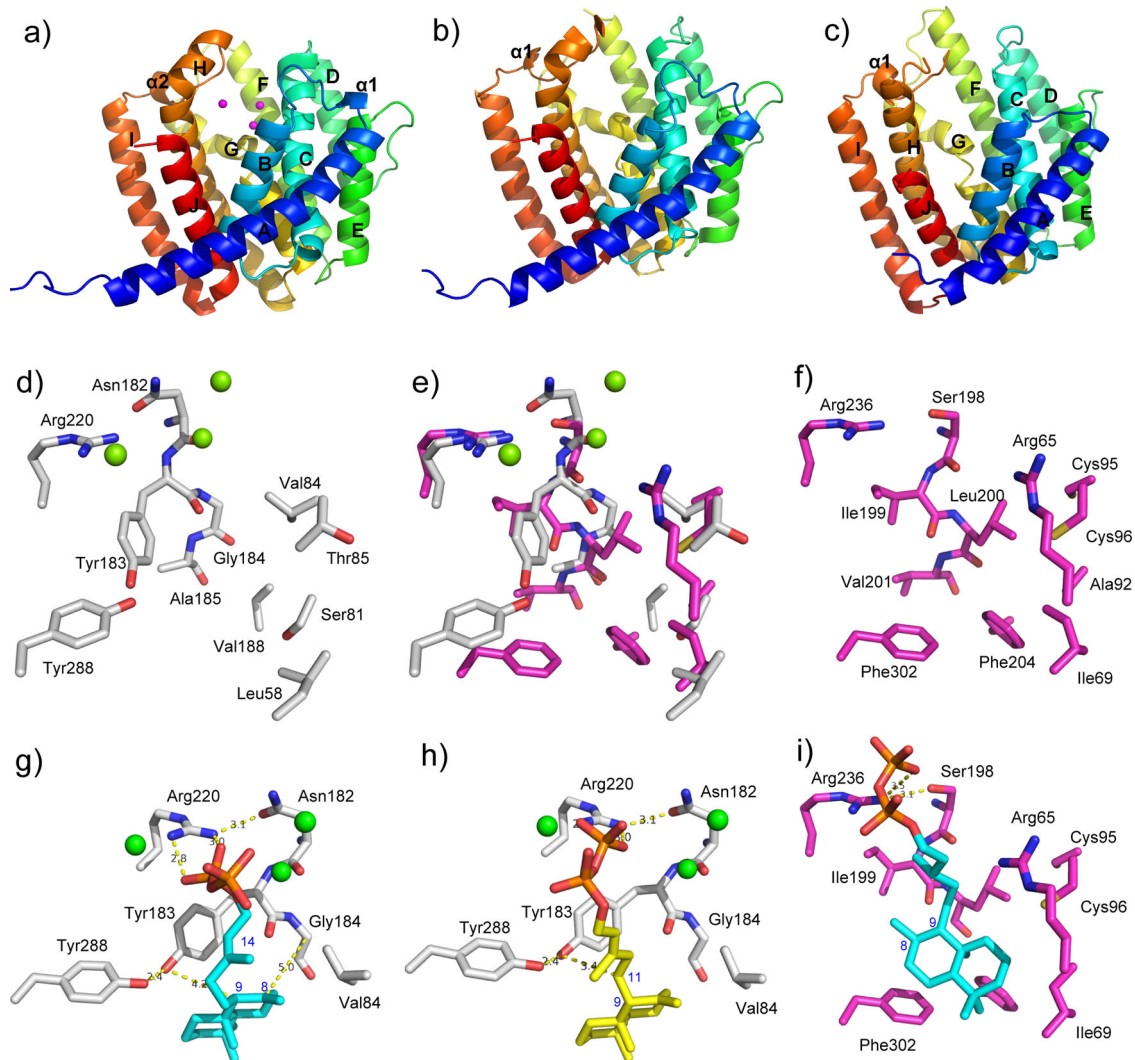

**Fig. 3 The crystal structures and active sites of Sat1646-Mg$^{2+}$, Sat1646, and Stt4548, and substrates modeling. a** The crystal structure of Sat1646-Mg$^{2+}$ (Mg$^{2+}$ ions in magenta spheres). **b** The crystal structure of Sat1646. **c** The crystal structure of Stt4548. **d** The active site of Sat1646-Mg$^{2+}$. **e** The superimposition of active sites of Sat1646-Mg$^{2+}$ and Stt4548. **f** The active site of Stt4548. **g** The modeling of CPP (cyan sticks) into the active site of Sat1646-Mg$^{2+}$. **h** The modeling of *syn*-CPP (yellow sticks) into the active site of Sat1646-Mg$^{2+}$. **i** The modeling of CPP (cyan sticks) into the active site of Stt4548. The structures in **a**–**c** are shown with coloring from blue on the N-terminus to red on the C-terminus. In **d**–**i**, the residues and Mg$^{2+}$ ions of Sat1646-Mg$^{2+}$ are shown in gray sticks and green spheres, respectively, and the residues of Stt4548 are shown in magenta sticks.

sequence and residues 22–313 were encoded by the *stt4548* sequence). Analysis of the crystallographic packing also shows that apo-Stt4548's N-terminal helix A interacts with helices B and J from an adjacent molecule (Fig. S11A). The apo-Stt4548 is superimposed well to Sat1646-Mg$^{2+}$, with an RMSD of 1.77 Å for 238 Cα atoms (Fig. S11B), and displays the canonical DDCAD$_{99–103}$ and NDVASHRRE$_{239–247}$ motifs characteristic of class I terpene synthases (Fig. S11C). Dali alignments for Sat1646-Mg$^{2+}$ and Stt4548 show that both of the DTSs are most structurally similar to LrdC, a class I DTS in (*E*)-biformene biosynthesis[21]. This is not surprising, since we now know that Sat1646, Stt4548, and LrdC each use the same substrate: CPP (Fig. S2B). However, we can infer that these enzymes employ distinct catalytic mechanisms because Sat1646-Mg$^{2+}$ and Stt4548 produce isopimara-8,15-dien (**1**) whereas LrdC produces (*E*)-biformene.

**Substrate modeling and site-directed mutagenesis reveals catalytic residues of Sat1646.** The apo-Sat1646 and Sat1646-Mg$^{2+}$ structures afford us the opportunities to identify and test putative

key catalytic residues in the active site. Detailed analyses of both structures suggested that Leu58 from helix B, Ser81, Val84, and Thr85 from helix C, Tyr183, Gly184, Ala185, and Val188 from helix G, and Tyr288 from helix J collectively define the active site (recall this is positioned below the three Mg$^{2+}$ ions) (Fig. 3d). Previous work has established that bacterial class I terpene synthases can utilize an induced-fit mechanism for substrate binding and ionization; this involves a so-called "effector triad" comprising sensor, linker, and effector residues that are located at the helix G1/2-break motif[22]. Consistently, we identified clear linker and effector residues in Sat1646: Asn182 and Tyr183 located at the helix G1/2-break motif. However, a distinction with Sat1646 is that its sensor residue Arg220 occurs in helix H rather than helix G, the sensor position of previously reported bacterial class I terpene synthases.

To investigate the catalytic mechanism of Sat1646, we modeled CPP into the active site of Sat1646-Mg$^{2+}$ by AutoDock. The diphosphate moiety of CPP, expectedly, is coordinated by the three Mg$^{2+}$ ions (Fig. 3g). The sensor Arg220 forms hydrogen bonds with the linker Asn182 and the diphosphate moiety, and

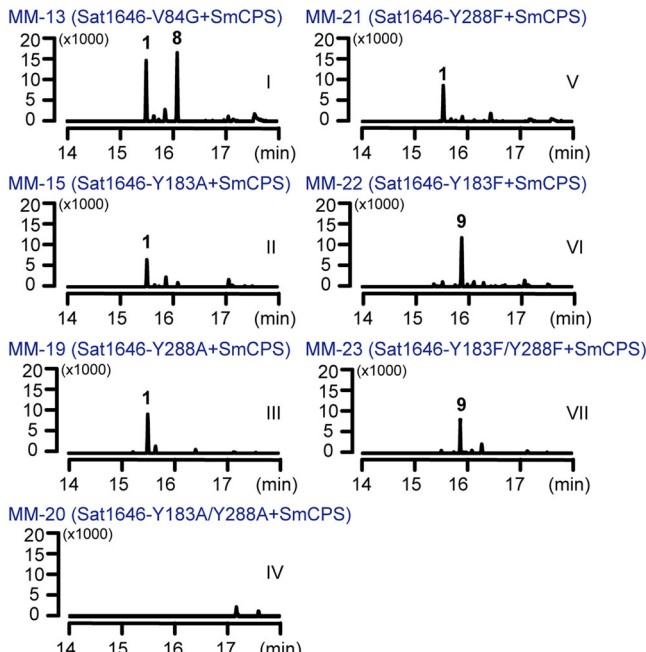

**Fig. 4 The GC–MS analysis (extracted ion chromatogram at *m/z* 272) of different engineered *E. coli* strains.** The *Y*-axes show the ion abundances.

the main chain carbonyl group of the effector Tyr183 points towards the 13/14 double bond of CPP (Fig. 3g); this conformation may facilitate departure of the diphosphate moiety to yield the allylic carbocation intermediate (**10**) and PP$_i$.[22]

The copalyl moiety of CPP is stuck between Tyr183-Tyr288 on one side and Val84 on the other side (Fig. 3g), and Tyr183 is close to C-8 of CPP (a distance of 4.5 Å). In light of previously proposed DTS catalysis mechanisms, we speculate that Tyr183 can stabilize the isopimara-15-en-8-yl⁺ intermediate (**11**) in the cyclization reaction based on π–cation interactions between the Tyr183's phenyl moiety and the C-8 carbocation.[23] When CPP adopts an 9S configuration, its C-9 is positioned close to the side chain hydroxy oxygen atom of Tyr183 (4.2 Å between C-9 and side chain hydroxy oxygen atom of Tyr183). This arrangement suggests that Tyr183 can function as the base for the deprotonation at CPP C-9. Further, it seems likely that Tyr288 functions in the activation of Tyr183 as a strong catalytic base: there is a hydrogen bond (2.4 Å) formed between the side-chain hydroxy oxygen atoms of Tyr183 and Tyr288 (Fig. 3g).

We next generated a series of Sat1646 variants with substitution mutations at candidate active site residues mentioned above, including L58A, S81A, V84G, T85A, Y183A, G184F, A185S, V188A, and Y288A. GC–MS analysis showed that six of the nine tested mutants produced **1** as the sole product, at yields similar to wild-type Sat1646 (Fig. S12). The exceptions were Sat1646$^{V84G}$, which produced **1** and new product **8** (Fig. 4, lane I), and both Sat1646$^{Y183A}$ and Sat1646$^{Y288A}$, which each produced decreased yields of **1** (Fig. 4, lanes II and III). Thus, residues Tyr183, Tyr288, and Val84 are important for Sat1646 function.

Follow-up work showed that a Sat1646$^{Y183A/Y288A}$ double mutant variant produced no diterpene product (Fig. 4, lane IV), and Sat1646$^{Y288F}$ variant could still produce **1** (albeit with a significantly decreased yield) (Fig. 4, lane V), supporting Tyr183-Tyr288 as a catalytic dyad of Sat1646. These results are consistent with our CPP modeling results: Tyr183 supports abstraction of the proton from C-9 during production of **1**, and we suspect that Tyr288 can function to activate Tyr183's side chain hydroxy group. Interestingly, it appears that Tyr288 may have a capacity

to (at least partially) compensate for the catalytic contribution of Tyr183: a small amount of **1** was detected in cells expressing the Sat1646$^{Y183A}$ variant (Fig. 4, lane II).

We also found that both Sat1646$^{Y183F}$ and Sat1646$^{Y183F/Y288F}$ variants blocked **1** production but enabled formation of a new product **9** (Fig. 4, lanes VI and VII). Compounds **8** and **9** were identified as the known compounds isopimara-7(8),15-diene and isopimara-8(14),15-diene, respectively (Fig. 1c), after large-scale fermentation, chromatographic purification, and spectroscopic analysis (Supplementary Information). The distinct double bond positions of **8** and **9** compared to **1** support that the Sat1646$^{V84G}$, Sat1646$^{Y183F}$, and Sat1646$^{Y183F/Y288F}$ variants employ alternative deprotonation mechanisms.

We also modeled *syn*-CPP into the active site of Sat1646-Mg²⁺ to investigate potential mechanisms for generating diverse products **2–7**. The binding *syn*-CPP at the Sat1646 active site is quite similar to that of CPP: *syn*-CPP's decalin skeleton assumes an almost identical position to that of CPP (Fig. 3h). However, the 9R configuration of *syn*-CPP positions the C-9 away from the Tyr183-Tyr288 catalytic dyad; this difference excludes deprotonation at C-9 by Tyr183-Tyr288 as proposed for the deprotonation of CPP. Such a difference may contribute to the alternative catalytic modes of Sat1646 when reacting with *syn*-CPP, generating diverse products **2–7** (Fig. S14).

**Substrate modeling and site-directed mutagenesis reveals distinct modes for CPP-binding and catalysis of Stt4548.** In addition to the canonical Mg²⁺-binding motifs characteristic of class I terpene synthase, Stt4548 also possesses a putative "effector triad" for substrate binding and ionization, comprising sensor Arg236, linker Ser198, and effector Ile199; these respectively correspond to Sat1646's Arg220, Asn182, and Tyr183 residues (Fig. 3e, f). Similar to Arg220 in Sat1646, Stt4548's Arg236 is located at helix H. Despite these similarities, Stt4548 has an active site below the Mg²⁺-binding motifs that is distinct from Sat1646. The counterparts of Sat1646's active site-defining residues Leu58, Ser81, Val84, Thr85, Tyr183, Gly184, Ala185, and Val188 are Ile69, Ala92, Cys95, Cys96, Ile199, Leu200, Val201, and Phe204 in Stt4548, respectively (Fig. 3e, f). The side chains of another two residues Phe302 and Arg65 in Stt4548 protrude into the active site center. Notably, the Sat1646's catalytic residue Tyr183 is replaced by Ile199 in Stt4548; Val84 is replaced by Cys95 in Stt4548; owing to lacking electron density, we did not build the Pro306 corresponding to Sat1646's Tyr288 in the structure of Stt4548 (Fig. 3e, f).

Analyzing the residues below the Mg²⁺-binding motifs in the active site suggested that three polar residues (Cys95, Cys96, and Arg65) in Stt4548 could act as bases for deprotonation at CPP's C-9. Modeling of CPP into the active site of Stt4548 showed that CPP's C-9 is positioned away from Cys95/Cys96 and Arg65 (Fig. 3i). Nevertheless, the model indicated that the decalin moiety of CPP is stabilized by hydrophobic interactions with Phe204, Phe302, Leu200, and Ile69, and there is a huge space around C-9 in the Stt4548 active site. The lack of residues around C-9 is similar to the lack of residues for deprotonation in the active site of PaFS in the biosynthesis of fusicoccadiene.[24] The PP$_i$ has been proposed as a general base for deprotonation in the catalysis of PaFS[24] and other terpene synthases such as TXS.[25] We propose that PP$_i$ may act as the catalytic base to abstract C-9 proton in the generation of **1** by Stt4548. Analysis of Stt4548 mutants showed that the Stt4548$^{C95A}$, Stt4548$^{C96A}$, and Stt4548$^{R65A}$ variants all produced **1** at similar yields as wild-type Stt4548, excluding direct catalytic contributions from these three polar residues (Fig. S13).

**Proposed catalytic mechanisms and comparisons with other DTSs.** Our assay data, substrate modeling, and site-directed

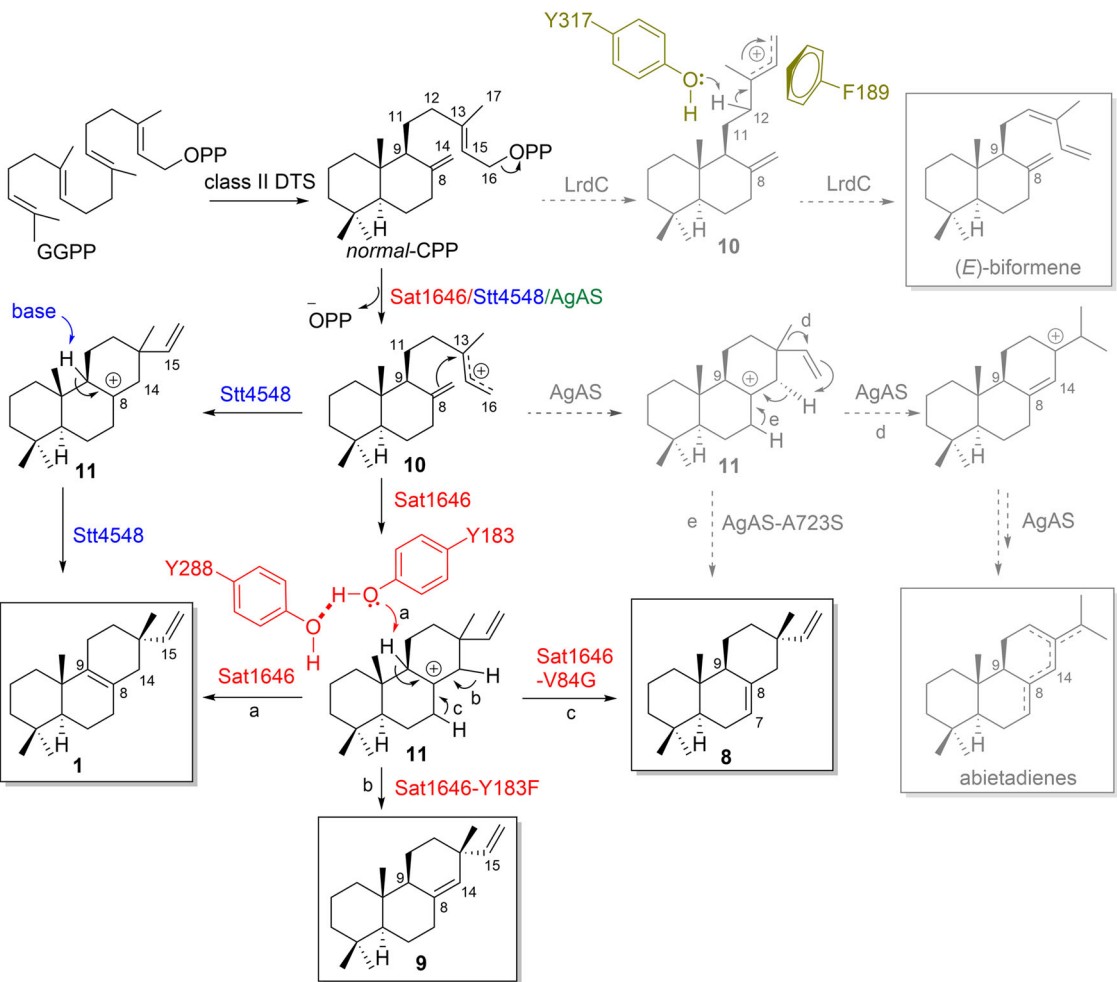

**Fig. 5 The comparison of proposed catalytic mechanisms of Sat1646/mutants (red) and Stt4548 (blue) with those of LrdC (magenta) and AgAS/ mutant (green).** The isolated products are shown in black squares. The biosynthetic pathways catalyzed by LrdC and AgAS/mutants are indicated with dashed arrows.

mutagenesis results support the following proposal for the catalytic mechanism used by Sat1646 to produce **1** (Fig. 5). During substrate (CPP) binding, three $Mg^{2+}$ ions coordinate the positioning of the diphosphate moiety into the active site, and the sensor residue Arg220 forms hydrogen bonds with both the diphosphate and the linker Asn182, thereby closing the active site; this would mirror a previously reported "induced-fit" mechanism for class I terpene synthases[22]. Subsequently, the effector residue Tyr183 provokes the departure of the diphosphate moiety to generate the allylic carbocation intermediate (**10**), whose cation charge is distributed among C-13/15/16. Cation-triggered cyclization then generates the isopimara-15-en-8-yl$^+$ intermediate (**11**), which is stabilized by π-cation interactions from Tyr183. The side chain hydroxy group of Tyr183, which is activated through hydrogen-bond interaction with the side chain hydroxy group of Tyr288, finally abstracts the proton from C-9 to generate **1** (Fig. 5).

In this proposed mechanism, CPP is stuck tightly between Tyr183-Tyr288 and Val84; this should support highly efficient deprotonation at C-9, helping to explain our observation of **1** as the sole product. This mechanism can also explain our observation that mutating Tyr183 and/or Val84, which would loosen or change the CPP binding as that in wild-type Sat1646 active site, enables alternative deprotonation reactions at C-14 or C-7, to respectively generate **8** and **9** (Fig. 5).

Eleven DTS structures have been reported to date (Fig. S2A). The catalytic dyad Tyr183-Tyr288 and Val84 constitute a unique active site in Sat1646, which has not been observed in the other 11 DTSs. Although dual tyrosine residues have been reported in the catalysis of Rv3378c, they do not interact with each other for a strong base activation but instead participate in hydroxylation at different positions of substrate[26]. Our comparative analysis focused on Sat1646, Stt4548, LrdC, and the α domain of AgAS; these all accept the same CPP substrate yet produce different labdane-related products (Fig. 5 and S2B)[27]. It has been shown that one mutation from Ala723 to Ser723 in AgAS could switch the enzyme from producing abietadienes to isopimara-7(8),15-diene (**8**) (Fig. 5)[19]. The crystal structure of AgAS supports that Ser723 stabilizes the isopimara-15-en-8-yl$^+$ intermediate (**11**) through a dipole–cation interaction, which can "short-circuit" the reaction to facilitate the production of **8**[28]. Structural comparison between Sat1646 and AgAS shows that the counterpart residue of Ala723 in AgAS is Gly184 in Sat1646 (Fig. 6a), whose Cα is only 5.0 Å from C-8 of CPP in our modeling (Fig. 3g). Although Gly184 in Sat1646 cannot provide the dipole–cation stabilization, the Tyr183 in Sat1646 can stabilize **11** by π–cation interactions to generate **1** (Fig. 3g).

The catalytic mechanism of LrdC has been well characterized, wherein two residues are essential for the generation of (E)-biformene from CPP: Phe189 stabilizes the allylic carbocation

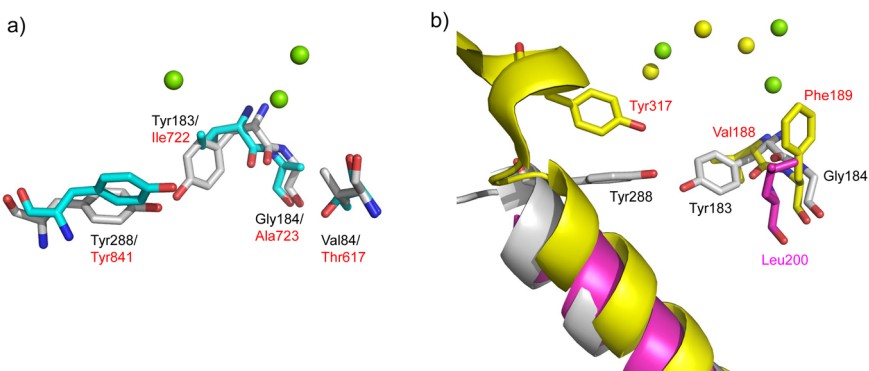

**Fig. 6 The key residues in the active sites of Sat1646-Mg$^{2+}$, Stt4548, LrdC, and AgAS. a** The comparison of key residues in the active sites of Sat1646-Mg$^{2+}$ (residues in gray and Mg$^{2+}$ ions in green spheres) and AgAS (residues in cyan). **b** The comparison of key residues in the active sites of Sat1646-Mg$^{2+}$ (residues in gray and Mg$^{2+}$ ions in green spheres), Stt4548 (residues in magenta) and LrdC (residues in yellow and Mg$^{2+}$ ions in yellow spheres).

intermediate (**10**) by π–cation interactions and Tyr317 abstracts C-12 proton to generate (*E*)-biformene (Fig. 5)[21]. Comparison of the active sites among LrdC, Sat1646, and Stt4548 indicated that LrdC's Phe189 is replaced by Gly184 in Sat1646 and by Leu200 in Stt4548 (Fig. 6b), suggesting that stabilization of **10** is improbable for Sat1646/Stt4548. The Tyr317 in LrdC has no corresponding residues at the similar positions in Sat1646 and Stt4548, because the C-terminal regions of Sat1646 and Stt4548 were not built due to the lack of electron densities (Fig. 6b). These differences could contribute to the different conversions of CPP to (*E*)-biformene or **1**. In summary, our proposed catalytic mechanisms and structural comparisons should be useful for DTS engineering and structure–function understanding to generate diverse diterpene skeletons.

## Discussion

Relative to the vast diversity of diterpenoids discovered from plants[29] and fungi[30], the extent of diterpenoids identified from bacteria is much less substantial. In the present study we identified two diterpenoid biosynthetic gene clusters—one from the marine strain *Salinispora* sp. PKU-MA00418 and the other from the soil strain *Streptomyces* sp. PKU-TA00600—based on a PCR-mediated genome mining method. The class I DTSs Sat1646 and Stt4548 from the two strains were characterized for their catalytic functions in the biosynthesis of isopimara-8,15-diene (**1**). Crystal structures of Sat1646 and Stt4548 were solved, revealing distinctions in the active sites of these two enzymes. Subsequently, both site-directed mutagenesis and substrate modeling identified a catalytic dyad comprising Tyr183 and Tyr288 in Sat1646.

For Sat1646, efficient production of **1** (as the sole product) from CPP involves the deprotonation by the Tyr183-Tyr288 dyad as well as the steric bulk of Val84. In contrast, binding of *syn*-CPP in Sat1646—which differs from CPP in its C-9 configuration—does not support efficient collaboration from these three residues for deprotonation at C-9. Rather, Sat1646 can catalyze deprotonation of *syn*-CPP at alternative positions, which supports distinct cyclization and rearrangement patterns en route to forming **2**–**7**. Stt4548 can also catalyze production of **1**, but this is achieved based on distinct interactions with CPP which are not shared with Sat1646, highlighting the varied strategies by DTSs in generating the same diterpene product.

To date, 11 DTSs have been structurally characterized and only limited details of the catalytic mechanisms of DTSs have been confirmed. Given that no structures of pimarane-type DTS enzymes have been solved previously, our structure of Sat1646 in the present study offers experimental support for the involvement of a unique catalytic dyad Tyr183-Tyr288 in isopimaradienes/

pimaradienes generation. It was known previously that LrdC and the α domain of AgAS accept the same substrate (CPP) as Sat1646/Stt4548; however, it is now clear that distinct carbocation stabilization mechanisms in their active sites drive their respective reactions towards formation of different products (iso-pimaradiene/pimaradiene, abietadiene or biformene). Beyond highlighting how active site plasticity can generate structurally diverse diterpene products from a single substrate, our insights about key residues and catalytic mechanisms should facilitate both rational engineering of DTSs and combinatorial biosynthesis to generate a variety of pimarane-type diterpenoids in the future.

## Methods

**General experimental procedures**. IR spectra were recorded with a NICOLET iS50 FT-IR spectrometer (Thermo Scientific, Waltham, MA, USA). NMR data were collected on a Bruker Avance-400FT and Avance-600 NMR spectrometers (Bruker Corporation, Billerica, MA, USA). GC–MS analyses were performed on an Agilent 7890A GC system connected to an Agilent 5975 mass spectrometer with a HP-5MS column (30 m × 250 μm × 0.25 μm, Agilent Technologies, Santa Clara, CA, USA). HREIMS spectrum was collected on a Hybrid Quadrupole-Orbitrap GC-MS/MS System (Thermo Scientific, Waltham, MA, USA) equipped with a TraceGOLD TG-5HT column (30 m × 250 μm × 0.1 μm, Thermo Scientific, Waltham, MA, USA). Size-exclusion chromatography was carried out with Sephadex LH-20 (GE Healthcare, Chicago, IL, USA) columns. HPLC analysis was performed on an Agilent 1260 series (Agilent Technologies, Santa Clara, CA, USA) with a C18 RP-column (Extend-C18, 250 × 4.6 mm, 5 μm, Agilent Technologies, Santa Clara, CA, USA). Semi-preparative HPLC was performed on an SSI 23201 system (Scientific Systems Inc., State College, PA, USA) with a YMC-Pack ODS-A column (250 × 10 mm, 5 μm, YMC CO., LTD. Shimogyo-ku, Kyoto, Japan). All fermentations were carried out in MQD-B1R shakers (Minquan Instrument Co., Ltd., Shanghai, China).

**PCR-mediated genome mining and phylogenetic analysis**. The genomic DNAs of all 1150 strains were extracted following standard protocols[31]. The forward primer (5′-TACGASACMGCSCGGMTGGT-3′) and the reverse primer (3′-ACCGTGCGSAGVGGCATGATGCG-5′), designed from the conserved nucleotide sequences of five *ent*-CPP synthases-encoding genes (*orf2, swt1.2, swt2.2, ptmT2 and ptnT2*)[12], and one *normal*-CPP synthase-encoding gene (*saCPS*)[13], were used in the PCR screening. A 20 μL of system consisting of 10 μL Easy Taq Polymerase (TransGen Biotech, Beijing, China), 1.6 μL of forward and reverse primers mixture (final concentration of 10 μM for each primer), 1.2 μL genomic DNA, 1.6 μL DMSO, and 4.0 μL sterilized H$_2$O, were used for PCR. The PCR program was performed with an initial denaturation at 95 °C for 5 min, followed by 35 cycles of denaturation at 95 °C for 30 s, annealing at 57 °C for 30 s and extension at 72 °C for 1 min, followed by incubation at 72 °C for 5 min. The PCR products were analyzed by agarose gel electrophoresis, recovered with a gel purification Kit (Vazyme Biotech Co., Ltd., Nanjing, China), and cloned into the pEASY-T1 vector (TransGen Biotech, Beijing, China) for sequencing.

To generate the phylogenetic analysis for strain PKU-MA00418 and PKU-TA00600, the genomic DNAs of PKU-MA00418 and PKU-TA00600 were extracted by using salting out method[31]. The 16S rRNA genes (GeneBank accession number MW550285 and MW550284) of PKU-MA00418 and PKU-TA00600 were amplified by PCR using the forward primer (5′-AGAGTTTGATCMTGGCTCAG-3′) and reverse primer (5′-TACGGYTACCTTGTTACGACTT-3′)[32]. Homologous genes were searched using BLAST on the NCBI website, and the phylogenetic tree

was generated with Mega 6.0 (Pennsylvania State University, State College, PA, USA) using the Neighbor-Joining algorithm.

**Genome sequencing and bioinformatics analysis of *sat* and *stt* clusters.** Genome sequencing was accomplished with the Illumina Hiseq technology by Shanghai Majorbio Bio-pharm Technology Co., Ltd. Biosynthetic gene clusters including the *sat* and *stt* clusters were detected and analyzed with antiSMASH[33]. The biosynthetic genes in *sat* and *stt* clusters were analyzed using ORFfinder and BLAST (Table S1)[34].

**Construction of engineered *E. coli* heterologous systems and site-directed mutagenesis.** The precursor biosynthetic genes *idi*, *dxr*, and *dxs* from *E. coli* were amplified by PCR and individually cloned into plasmid pRSFDuet-1 with the Gibson assembly method to generate the plasmid pMM-1, and the GGPP synthase PtmT4-encoding gene was synthesized and cloned into plasmid pCDFDuet-1 with the same method to generate pMM-2[14]. The genes *sat1645* and *sat1646* were amplified by PCR and cloned into pMM-2 to generate pMM-3. The two plasmids pMM-1 and pMM-3 were cotransformed into *E. coli* BL21 (DE3) (TransGene Biotech, Beijing, China) to afford MM-1, and fermentation followed by GC–MS analysis showed that no diterpenoid was produced in the fermentation of MM-1. We tested the expression of *sat1645* and *sat1646* individually in *E. coli* BL21 (DE3), affording overproduction of soluble Sat1646 but no soluble Sat1645. Thus, Sat1645 may be insoluble in MM-1, leading to no substrate produced for Sat1646. Previous combination of *saDTS* and *saCPS* in engineered *E. coli* heterologous system also gave low production of diterpenoid, and *saCPS* (homologue of *sat1645*) was replaced by other *normal*-CPP synthase-encoding gene for diterpenoids production. We then synthesized *smCPS*, which encodes another *normal*-CPP synthase from *Salvia miltiorrhiza*, to replace *sat1645* in pMM-3 to generate pMM-4. Cotransformation of pMM-1 and pMM-4 into *E. coli* BL21 (DE3) generated MM-2. We also synthesized *osCPS4* to replace *smCPS* in pMM-4 to generate pMM-5. Cotransformation of pMM-1 and pMM-5 into *E. coli* BL21 (DE3) generates MM-3. To investigate what diterpene is produced by Stt4542 and Stt4548, *stt4542* and *stt4548* were amplified by PCR and cloned into pMM-2 to generate pMM-6. Cotransformation of pMM-1 and pMM-6 into *E. coli* BL21 (DE3) generated MM-4, producing no diterpenoid product based on GC–MS analysis. We then tested the expression of *stt4542* and *stt4548* individually in *E. coli* BL21 (DE3), affording overproduction of soluble Stt4548 but no soluble Stt4542. Thus, similar to Sat1645, Stt4542 may be insoluble in MM-4, leading to no substrate produced for Stt4548. To explore the possible substrate of Stt4548, *stt4548* was cloned together with each of class II DTS-encoding genes including *smCPS* (encoding *normal*-CPP synthase), *osCPS4* (encoding *syn*-CPP synthase), *ptmT2* (encoding *ent*-CPP synthase), *haur_2145* (encoding KPP synthase), *kgTPS* (encoding *syn*-KPP synthase), and *mtHPS* (encoding TPP synthase), into pMM-2 to generate pMM-7 to pMM-12, respectively. Cotransformation of pMM-1 and each of pMM-7 to pMM-12 into *E. coli* BL21 (DE3) generated MM-5 to MM-10, respectively. The site-directed mutagenesis of *sat1646* or *stt4548*, using pMM-4 or pMM-7 as the template, were performed via PCR using primers containing the mutated sequences (Table S2), generating plasmid pMM-13 to pMM-29.

**The GC–MS analysis and purification of diterpenes.** Strains were cultivated in LB medium supplemented with 30 μg/mL kanamycin and 50 μg/mL spectinomycin for 8 h to afford seed cultures, which were inoculated into TB medium (tryptone 12 g, yeast extract 24 g, glycerol 4 mL, $KH_2PO_4$ 2.31 g, $K_2HPO_4$ 12.54 g in 1 L distilled $H_2O$) with 30 μg/mL kanamycin and 50 μg/mL spectinomycin. The fermentation was continued at 37 °C, 220 rpm until an $OD_{600}$ of 0.8 was reached, then the culture was cooled down to 18 °C for isopropyl β-D-thiogalactopyranoside (IPTG, final concentration: 0.5 mM) induction. The fermentation was continued for additional 72 h at 18 °C. This fermentation broths was extracted with an equal volume of hexane, and the hexane extracts were dried under vacuum with a rotary evaporator. For GC–MS analysis, a small part of the hexane extracts was dissolved in hexane and subjected to GC–MS analysis. Samples (1 μL) were injected in splitless mode at 50 °C. After being held at 50 °C for 3 min, the oven temperature was raised to 300 °C with a rate of 14 °C/min, then held for another 3 min. MS data was collected starting after 4 min.

For purification of diterpenes, large-scale fermentations were carried out as the same above. The hexane extracts were loaded to Sephadex LH-20 chromatography eluted with petroleum ether/chloroform/methanol (5: 5: 1) to yield a series of fractions. Fractions containing diterpenes were monitored by TLC and dried under a gentle stream of $N_2$, and purified by semi-preparative HPLC eluted with 100% acetonitrile (0–60 min) at a flow rate of 2.0 mL/min, under the UV detection at 210 nm.

**Structural elucidation of diterpenes.** Compound **2** was isolated as a colorless oil. HREIMS analysis afforded an $M^+$ ion at *m/z* 272.24976, giving the molecular formula of **2** as $C_{20}H_{32}$. The $^1H$ and $^{13}C$ NMR spectra of **2** showed similar signals attributed to the A/B ring system of *syn*-isopimara-7(8),15-diene (**3**), but the left carbons (C-11 to C-17) and protons (H-11 to H-17) gave different chemical shifts and coupling systems from those of **3**. The signals attributed to the mono-substituted terminal double bond in **3** were replaced by signals attributed to a disubstituted terminal double bond at $\delta_H$ 4.70 (2H, m, H-14), $\delta_C$ 155.1 (C-13) and

108.7 (C-14) in **2**, and the singlet signal attributed to the methyl group at $\delta_H$ 1.00 (3H, s, H-17) in **3** was replaced by the doublet signal attributed to the methyl group at $\delta_H$ 1.05 (3H, d, *J* = 6.0 Hz, H-17) in **2**. The presence of a seven-membered ring fused to ring B on C-8/9 in **2** was confirmed based on the COSY correlations from H-11 to H-12, H-16 to H-15 and H-17, and the HMBC correlations from H-14 to C-12 and C-15, H-11/12/16/15/17 to C-13, H-11/12/16 to C-9, and H-11/16/15 to C-8, and H-7 to C-16. Detailed analyses of the DEPT, COSY, HSQC, and HMBC established the planar structure of **2**, which possesses a "6/6/7" ring system. The NOESY correlation between H-9 and H-15 identified that the two protons were on the same side of the seven-membered ring. Since the absolute configurations on C-5/9/10 of **2** have been known from the substrate *syn*-CPP, the absolute configuration on C-15 of **2** was established as 15*R*.

*Compound* **2**: Colorless oil; TLC (petroleum ether): $R_f$ = 0.78; [α]$^{25}_D$ +26.67 (*c* 0.05, CHCl₃); UV (CHCl₃) $\lambda_{max}$ (log ε) 208 (3.81) nm; IR (diamond ATR) $\nu_{max}$: 3404, 2923, 2867, 1714, 1460, 1377, 1091, 887 cm⁻¹; $^1H$ NMR (600 MHz) data, Table S5; $^{13}C$ NMR (150 MHz) data, Table S6; HREIMS *m/z* 272.24976 M⁺ (calcd for $C_{20}H_{32}$, 272.24985).

The structures of **1** and **3–9** were elucidated based on comprehensive analyses of their NMR and MS spectra. Compound **1** was isolated as a colorless oil. EIMS analysis afforded an M⁺ ion at *m/z* 272.2. Compound **1** was identified as isopimara-8(9),15-diene based on the comparison of its $^1H$ NMR data (Table S5) and $^{13}C$ NMR data (Table S6) with authentic data published before[13]. Compound **3** was isolated as a colorless oil. EIMS analysis afforded an M⁺ ion at *m/z* 272.2. Compound **3** was identified as *syn*-isopimara-7(8),15-diene based on the comparison of its $^1H$ NMR data (Table S5) and $^{13}C$ NMR data (Table S6) with authentic data published before[35]. Crystal structure of **3** was solved, clarifying the absolute configuration. Compound **4** was isolated as a colorless oil. EIMS analysis afforded an M⁺ ion at *m/z* 272.2. Compound **4** was identified as 8β-isopimara-9(11),15-diene based on the comparison of its $^1H$ NMR data (Table S5) and $^{13}C$ NMR data (Table S6) with authentic data published before[16]. Crystal structure of **4** was solved, clarifying the absolute configuration. Compound **5** was isolated as a colorless oil. EIMS analysis afforded an M⁺ ion at *m/z* 272.2. Compound **5** was identified as 8β-pimara-9(11),15-diene based on the comparison of its $^1H$ NMR data (Table S5) and $^{13}C$ NMR data (Table S6) with authentic data published before[36]. Crystal structure of **5** was solved, clarifying the absolute configuration. Compound **6** was isolated as a colorless oil. EIMS analysis afforded an M⁺ ion at *m/z* 272.2. Compound **6** was identified as *syn*-stemod-13(17)-ene based on the comparison of its $^1H$ NMR data (Table S5) and $^{13}C$ NMR data (Table S6) with authentic data published before[37]. Crystal structure of **6** was solved, clarifying the absolute configuration. Compound **7** was isolated as a colorless oil. EIMS analysis afforded an M⁺ ion at *m/z* 272.2. Compound **7** was identified as *syn*-pimara-7(8),15-diene based on the comparison of its $^1H$ NMR data (Table S5) and $^{13}C$ NMR data (Table S6) with authentic data published before[38]. Compound **8** was isolated as a colorless oil. EIMS analysis afforded an M⁺ ion at *m/z* 272.2. Compound **8** was identified as isopimara-7(8),15-diene based on the comparison of its $^1H$ NMR data (Table S5) and $^{13}C$ NMR data (Table S6) with authentic data published before[39,40]. Compound **9** was isolated as a colorless oil. EIMS analysis afforded an M⁺ ion at *m/z* 272.2. Compound **9** was identified as isopimara-8(14),15-diene based on the comparison of its $^1H$ NMR data (Table S5) and $^{13}C$ NMR data (Table S6) with authentic data published before[41]. The absolute configuration assignments of **2–7** were also benefitted by the structure of *syn*-CPP, whose absolute configurations have already been set from the catalysis of the class II DTS OsCPS4. The NMR spectra of **1–9** are presented in Supplementary Figs. S15–S30.

**X-ray crystallographic analysis of diterpenes.** Crystals were selected and loaded onto a XtaLAB Synergy R, HyPix diffractometer. Crystals were kept at 100 K during data collection. The structures were solved with Olex2[42]/ShelXS[43] programs by Direct Methods, and refined with the ShelXL[44] package by Least Squares minimization. The crystallographic data have been deposited at the Cambridge Crystallographic Data Centre with deposition numbers CCDC 2054936 (**3**), 2054939 (**4**), 2054937 (**5**), 2054940 (**6**).

*Crystallographic Data for* **3**: Suitable crystals were obtained from **3** in an chloroform solution, $C_{20}H_{32}$ (*M* = 272.45 g/mol): orthorhombic, space group P2₁2₁2₁ (no. 19), *a* = 7.40170(10) Å, *b* = 10.13280(10) Å, *c* = 22.4655(3) Å, *V* = 1684.91(4) Å³, *Z* = 4, *T* = 100.01(10) K, μ(CuKα) = 0.432 mm⁻¹, *Dcalc* = 1.074 g/cm³, 31676 reflections measured (7.87° ≤ 2Θ ≤ 144.09°), 3280 unique ($R_{int}$ = 0.0443, $R_{sigma}$ = 0.0232) which were used in all calculations. The final $R_1$ was 0.0372 (I > 2σ(I)) and $wR_2$ was 0.0937. The Flack parameter was 0.0(3).

*Crystallographic Data for* **4**: Suitable crystals were obtained from **4** in an hexane solution, $C_{20}H_{32}$ (*M* = 272.45 g/mol): orthorhombic, space group P2₁2₁2₁ (no. 19), *a* = 10.69593(9) Å, *b* = 12.38293(8) Å, *c* = 12.49611(8) Å, *V* = 1655.07(2) Å³, *Z* = 4, *T* = 100 K, μ(CuKα) = 0.440 mm⁻¹, *Dcalc* = 1.093 g/cm³, 31803 reflections measured (10.056° ≤ 2Θ ≤ 149.928°), 3340 unique ($R_{int}$ = 0.0387, $R_{sigma}$ = 0.0167) which were used in all calculations. The final $R_1$ was 0.0346 (I > 2σ(I)) and $wR_2$ was 0.0887. The Flack parameter was 0.0(2).

*Crystallographic Data for* **5**: Suitable crystals were obtained from **5** in an chloroform solution, $C_{20}H_{32}$ (*M* = 272.45 g/mol): orthorhombic, space group P2₁2₁2₁ (no. 19), *a* = 6.86830(10) Å, *b* = 12.18470(10) Å, *c* = 19.9030(2) Å, *V* = 1665.65(3) Å³, *Z* = 4, *T* = 100 K, μ(CuKα) = 0.437 mm⁻¹, *Dcalc* = 1.078 g/cm³, 32711 reflections measured (8.508° ≤ 2Θ ≤ 133.134°), 2932 unique ($R_{int}$ = 0.0444, $R_{sigma}$ = 0.0179)

which were used in all calculations. The final $R_1$ was 0.0295 (I > 2σ(I)) and $wR_2$ was 0.0769. The Flack parameter was 0.0(3).

*Crystallographic Data for 6*: Suitable crystals were obtained from **6** in an chloroform/methanol (5:1) solution, $C_{20}H_{32}$ (M = 272.45 g/mol): monoclinic, space group P2₁ (no. 4), a = 10.69420(10) Å, b = 7.61100(10) Å, c = 20.5454(3) Å, β = 102.4630(10)°, V = 1632.86(4) Å³, Z = 4, T = 100.00(10) K, μ(CuKα) = 0.446 mm⁻¹, Dcalc = 1.108 g/cm³, 26938 reflections measured (8.468° ≤ 2Θ ≤ 150.574°), 6488 unique (Rint = 0.0386, Rsigma = 0.0273) which were used in all calculations. The final $R_1$ was 0.0377 (I > 2σ(I)) and $wR_2$ was 0.1013. The Flack parameter was 0.0(3).

**Gene expression and purification of Sat1646 and Stt4548**. The DNA fragments of *sat1646* and *stt4548* were amplified by PCR from the genomic DNAs of *Salinispora* sp. PKU-MA00418 and *Streptomyces* sp. PKU-TA00600 with primers Sat1646-pMCSG19-F, Sat1646-pMCSG19-R, Stt4548-pET28a-F (*Nde*I site), and Stt4548-pET28a-R (*Eco*RI site) (Table S2). The purified PCR product of *sat1646* was introduced into pMCSG19 expression vector through ligation-independent cloning (LIC)[45], generating plasmid pMM-30. The purified PCR product of *stt4548* was digested with *Nde*I and *Eco*RI and ligated into pET28a vector with T4 DNA ligase, generating plasmid pMM-31.

For purification of Stt4548 for enzyme activity assays, the plasmid pMM-31 was transformed into *E. coli* BL21 (DE3) (TransGen Biotech, Beijing, China). The positive transformants (MM-29) were cultivated in 500 mL lysogeny broth (LB) with 30 μg/mL kanamycin at 37 °C, 220 rpm until an OD₆₀₀ of 0.6 was reached. IPTG (final concentration: 0.2 mM) was added for induction at 18 °C, and the fermentation was continued at 18 °C, 220 rpm for 18 h. Cells were harvested by centrifugation (4000 rpm, 10 min at 4 °C) and resuspended in cold lysis buffer (100 mM Tris-HCl, 300 mM NaCl, 15 mM imidazole, 10% glycerol, pH 8.0). The cells were lysed by sonication and the debris was removed by centrifugation (13,000 rpm, 50 min at 4 °C) and the supernatant was filtered by 0.45 μm membrane. The Stt4548 protein in the supernatant was purified with a 5 mL Histrap HP column (GE Healthcare, Chicago, IL, USA) on a ÄKTA chromatography system (GE Healthcare Life Sciences, Chicago, IL, USA). The concentration of Stt4548 was determined at 280 nm with a NanoDrop 2000C spectrophotometer (Thermo Scientific, Waltham, MA, USA).

For protein crystallization, the production of selenomethionine (SeMet)-labeled Sat1646 and Stt4548 were performed according to standard protocol[46]. The plasmid pMM-30 or pMM-31 was transformed into *E. coli* BL21 (DE3) and cultivated in 1 L of enriched M9 minimal medium[14] supplemented with 100 μg/mL ampicillin (for pMM-30) or 30 μg/mL kanamycin (for pMM-31) at 37 °C, 220 rpm until an OD₆₀₀ of 1.0 was reached. Inhibitory amino acids (50 mg/mL each of L-valine, L-isoleucine, L-leucine, 100 mg/mL each of L-lysine, L-threonine, L-phenylalanine) and 90 mg/L of L-SeMet were added to the culture for 15 min at 37 °C, 220 rpm. After the culture was cooled down to 18 °C, 0.2 mM IPTG were added. The fermentation was continued at 220 rpm for 20 h. Cells were harvested by centrifugation (4000 rpm, 10 min at 4 °C) and resuspended in cold lysis buffer (100 mM Tris-HCl, 300 mM NaCl, 15 mM imidazole, 10% glycerol, pH 8.0). The cells were lysed by sonication and the debris was removed by centrifugation (13,000 rpm, 50 min at 4 °C). The Se-Sat1646 or Se-Stt4548 in the supernatant was purified with a 5 mL Histrap HP column on a ÄKTA chromatographic system. The SeMet-labeled Sat1646 was incubated with TEV protease in 50 mM NaH₂PO₄, 150 mM NaCl, pH 8.0, at 16 °C for 10 h to remove the His₆ tag and MBP. After the digestion by TEV protease, the SeMet-labeled Sat1646 was purified with the 5 mL Histrap HP column again (protein in the flow-through part). The protein concentrations were determined at 280 nm with a NanoDrop 2000C spectrophotometer, and the homogeneities of Sat1646 and Stt4548 were analyzed by SDS-PAGE (Fig. S8).

**Crystallization, data collection, and structural elucidation of Sat1646 and Stt4548**. The SeMet-labeled Sat1646 (apo-Sat1646), SeMet-labeled Stt4548, and SeMet-labeled Sat1646-Mg²⁺ complex (incubated with 5 mM MgCl₂ and 10 mM PPi) were concentrated to 20 mg/mL for crystallization. With hanging-drop vapor-diffusion method at 16 °C, the crystals of SeMet-labeled Sat1646 and SeMet-labeled Sat1646-Mg²⁺ complex were obtained in screening kit Index-36 (Hampton Research) that contains 15% Tacsimate (pH 7.0), 100 mM HEPES (pH 7.0), and 2% w/v PEG 3350. The crystals of SeMet-labeled Stt4548 were obtained in screening kit Index-30 (Hampton Research) that contains 100 mM NaCl, 100 mM BIS-TRIS (pH 6.5), 1.5 M (NH₄)₂SO₄. X-ray diffraction data was collected at beamline BL17U1 at Shanghai Synchrotron Radiation Facility (SSRF)[47]. Diffraction data was first processed with HKL-2000[48], and phases were solved with PHENIX. Autosol[49,50]. Initial model was built with PHENIX. Autobuild and refined with PHENIX. Refine[51]. Final model was established with multiple rounds of manual modeling with Coot[52] and refinements with Refmac5 in CCP4[53] and PHENIX. Refine.

**Substrate modeling**. Substrate modeling was performed using AutoDock 4.2.27[54]. Before the docking, water molecules were removed from Sat1646 and Stt4548. AutoDockTools 1.5.6 was used to prepare the substrates *syn*-CPP and CPP, macromolecules Sat1646, Stt4548, and grid map files. The default parameters were used to set the torsion constraints for substrates *syn*-CPP and CPP, and charges and hydrogen atoms were added to Sat1646 and Stt4548 proteins. For the parameters of Sat1646 with *syn*-CPP and CPP, a grid box of 32 × 40 × 40 nm³ (x × y × z) with the search space center of

x = −8.64, y = −31.495, and z = −13.048 was created (grid spacing: 0.375 Å). For the parameters of Stt4548 with CPP, a grid box of 40 × 40 × 40 nm³ (x × y × z) with the search space center of x = −14.494, y = −53.96, and z = −9.406 was created (grid spacing: 0.375 Å). All dockings were performed using a Lamarckian genetic algorithm (LGA). The search parameters were used with default value.

**Reporting summary**. Further information on research design is available in the Nature Research Reporting Summary linked to this article.

## Data availablity
Data supporting the findings of this study are available within the manuscript and its Supplementary Information files. The crystal structures of Sat1646-Mg²⁺, Sat1646, and Stt4548 are accessible via PDB IDs of 7E4N, 7E4O, and 7E4M, respectively, at http://www.rcsb.org. Validation reports for the protein structures are available in the submitted Supplementary Data 1–3. The X-ray crystallographic coordinates for compounds **3**–**6** reported in this article have been deposited at the Cambridge Crystallographic Data Centre (CCDC), under deposition number CCDC 2054936 (**3**), 2054939 (**4**), 2054937 (**5**), 2054940 (**6**). The CIF files for these small molecule structures are submitted as Supplementary Data 4–7. These data can be obtained free of charge from The Cambridge Crystallographic Data Centre via www.ccdc.cam.ac.uk/data_request/cif.

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

## Acknowledgements

We thank Haiyan Tao of the State Key Laboratory of Natural and Biomimetic Drugs, School of Pharmaceutical Sciences, Peking University, for GC–MS analyses and IR spectra collection. We thank Qin Li and Fen Liu of the State Key Laboratory of Natural and Biomimetic Drugs, School of Pharmaceutical Sciences, Peking University, for NMR data collection. We thank the staffs from BL17U1/BL19U1 beamlines of National Facility for Protein Science in Shanghai (NFPS) at Shanghai Synchrotron Radiation Facility[47,55], for assistance during data collection. This research was supported in part by the National Key Research and Development Program of China (2019YFC0312502), the National Natural Science Foundation of China (grant numbers 22077007, 21877002, 81991525, 22107007, 81673332), the key project at central government level: the ability establishment of sustainable use for valuable Chinese medicine resources (2060302-1903-03), and China Postdoctoral Science Foundation (2019M660362).

## Author contributions

B. X. and J. Y. perform most investigations and contributed equally; C. C., X. M., Q. X., A. L., Y. G., Z. W., and T. L. help the investigation; H. J. and F. Y. provide resources for crystal structures elucidation; J. G., L. H., D. Y., and M. M. for formal analysis; M. M. for supervision and funding acquisition; D. Y. and M. M. for conceptualization; B. X. and J. Y. for writing-original draft; M. M. for writing-review and editing.

## Competing interests

The authors declare no competing interests.
