## [Peer Review File · Communications Chemistry]

Reviewers' comments:

Reviewer #1 (Remarks to the Author):

In the manuscript by Xing et al., reported their characterization of terpene cyclases from two biosynthetic pathways identified from their genome mining efforts. There are an extensive amount of data, including genome mining, assembly of gene clusters, heterogenous expression, product characterization, crystallographic characterization of two terpene cyclases, and structure-guided site-directed mutagenesis to provide mechanistic information regarding the reaction mechanisms. It is a lot of work and the results are interesting. Therefore, it is worthwhile to publish on Nature Communications.

Overall, there are an extensive amount of data. I reviewed the previous version of this manuscript submitted to Proc. Natl. Acad. Sci. In the previous version, there are way too much data included, which leads to a nearly 150 pages manuscript (main text + SI). In this draft, the authors have investigated extensive amount of efforts rewriting the manuscript, which makes it much easier to follow. In addition, the authors have taken all of my previous comments into consideration. Therefore, I support publishing it. The authors might need to ask some native speakers to polish the language a little bit. There are some typos.

Reviewer #2 (Remarks to the Author):

The authors have obtained two type I diterpene synthases for pimaranes, while clustered type II enzymes could not be obtained in soluble form. Therefore, known type II enzymes were used to substitute for them. Besides CPP (yielding isopimaradiene) also syn-CPP was accepted by one of the characterised type I enzymes, yielding six more products. Crystal structures of the enzymes were obtained, allowing for site-directed mutagenesis of active site residues, which gave two more products arising by alternative deprotonations of the latest intermediate. This ultimately led to a mechanistic model for the terpene cyclisation. The work is well explained and the manuscript is nicely written. Overall, this work may become acceptable, but some major points need improvement.

1. The discussion about the genome mining approach should be changed. The authors used degenerate primers in PCRs with genomic DNA of >1000 bacteria, resulting in only two hits. These turned out to be very close homologs of the genes that were the basis for the degenerate primers. Also the diterpene product was the same as reported previously. Why was not simply a synthetic gene for SaDTS used here? If any conclusion can be drawn from these data, then maybe that the target gene is not very widespread among bacteria.
2. The excessive self-citations (references 11 – 16) should be removed, as the content of this work is not discussed. These references are only added to a rather empty phrase („As part of our long-term research of natural product discovery and biosynthesis...“).
3. „nor-CPS“ should be changed. The prefix „nor“ means degraded by one carbon. The authors mean „normal“ CPP, but this does not need to be specified. Just CPP is enough.
4. It should be stated more clearly, which compounds are known and which are new. It seems that compound 2 is new, the others are known. For the new compound 2 full analytical data must be given not only NMR data. Please add IR, UV and HRMS data and, most importantly (!), the optical rotation. This is very important, because researchers can correlate to this, in case this compound is isolated again. As the absolute configuration of 2 is known, such a correlation can become highly

interesting in the future and will save a lot of unnecessary work, if 2 is isolated again.

5. Line 129: The absolute configuration of the diterpenes 3 – 6 was determined by X-ray. Actually, the observed absolute configurations are already set by the (known) type II diterpene synthase OsCPS4 that was used for the first enzymatic step. The opposite absolute configurations would not be expected. This should at least be discussed. This is only done for compound 2, hidden in the experimental part.

6. Several mutations of active site residues did not change the productivity of the enzyme, while three changes did. The authors conclude that these residues are involved in catalysis (implying the others are not). The interpretation should be done here more carefully. What means „involved in catalysis“? Direct interaction (by covalent bonds, hydrogen bonds, non-polar interaction)? For valine a participation in catalysis is difficult to understand. On the other hand, isn't the whole enzyme to be seen as „the catalyst“? In type I terpene cyclisations the enzyme is usually considered to function by ionisation of the substrate, while non-polar active site residues force the substrate in a reactive conformation for the cyclisation cascade that happens more or less independent of the enzyme. Maybe certain residues are relevant for cation stabilisations. The exchange of valine against alanine will likely only decrease the steric bulk in the active site, allowing the substrate to shift a bit within the active site, leading to different deprotonations. A direct involvement in catalysis is not expected.

7. Line 344: Reference 10 seems to be wrong here. The authors discuss diterpenoids from plants and fungi, but ref. 10 is from a streptomycete (bacterium).

8. SI Tables S5 and S6: references to previously published NMR data should be given, if available.

August 30th, 2021

Reviewer-1

Remarks to the Author:

In the manuscript by Xing et al., reported their characterization of terpene cyclases from two biosynthetic pathways identified from their genome mining efforts. There are an extensive amount of data, including genome mining, assembly of gene clusters, heterogenous expression, product characterization, crystallographic characterization of two terpene cyclases, and structure-guided site-directed mutagenesis to provide mechanistic information regarding the reaction mechanisms. It is a lot of work and the results are interesting. Therefore, it is worthwhile to publish on Nature Communications.

Overall, there are an extensive amount of data. I reviewed the previous version of this manuscript submitted to Proc. Natl. Acad. Sci. In the previous version, there are way too much data included, which leads to a nearly 150 pages manuscript (main text + SI). In this draft, the authors have investigated extensive amount of efforts rewriting the manuscript, which makes it much easier to follow. In addition, the authors have taken all of my previous comments into consideration. Therefore, I support publishing it. The authors might need to ask some native speakers to polish the language a little bit. There are some typos.

Response: We very much thank the reviewer's excellent and careful review! Indeed, there are too much data and unnecessary experimental descriptions in our previous version of manuscript and Supplementary Information. According to each of your comments, we substantially revised the manuscript and Supplementary Information, which had greatly improved the manuscript/story. For the new concerns about the language and typos, we have polished the language and corrected all the typos after checking the whole manuscript carefully. Our modifications are highlighted in red. We very much appreciate your professional comments and supportive guidance. Many thanks!

Reviewer-2

Remarks to the Author:

The authors have obtained two type I diterpene synthases for pimaranes, while clustered type II enzymes

could not be obtained in soluble form. Therefore, known type II enzymes were used to substitute for them. Besides CPP (yielding isopimaradiene) also syn-CPP was accepted by one of the characterised type I enzymes, yielding six more products. Crystal structures of the enzymes were obtained, allowing for site-directed mutagenesis of active site residues, which gave two more products arising by alternative deprotonations of the latest intermediate. This ultimately led to a mechanistic model for the terpene cyclisation. The work is well explained and the manuscript is nicely written. Overall, this work may become acceptable, but some major points need improvement.

Response: We thank the reviewer for the encouraging comments and for the professional suggestions/corrections about our study. According to each of your comments, we have revised the manuscript and Supplementary Information (modifications highlighted in red) and provided full point-by-point responses below:

(1) *The discussion about the genome mining approach should be changed. The authors used degenerate primers in PCRs with genomic DNA of >1000 bacteria, resulting in only two hits. These turned out to be very close homologs of the genes that were the basis for the degenerate primers. Also the diterpene product was the same as reported previously. Why was not simply a synthetic gene for SaDTS used here? If any conclusion can be drawn from these data, then maybe that the target gene is not very widespread among bacteria.*

Response: We thank the reviewer's insightful question and discussion. We agree that the degenerate primers we used may be very specific to some group of DTS-encoding genes, and the target genes from this screening may not be widespread among the 1150 bacteria. To make this important conclusion, we have added the sentence of "Only two hits (including the *sat* cluster with high homology with the known *terp1*) were discovered from this PCR screening, suggesting the target genes screened from our specific degenerate primers may not be widespread among the 1150 bacteria." into the genome mining part. We prefer to retain this genome mining approach in the text, because we indeed made great effort in this genome mining and we want to provide a reference for others who would consider a similar diterpenoid discovery method.

(2) *The excessive self-citations (references 11 – 16) should be removed, as the content of this work is not discussed. These references are only added to a rather empty phrase („As part of our long-term research of natural product discovery and biosynthesis...“).*

Response: We have removed the self-citations (Ref. 11-16) and re-organized the references.

(3) "nor-CPS" should be changed. The prefix "nor" means degraded by one carbon. The authors mean "normal" CPP, but this does not need to be specified. Just CPP is enough.

Response: We thank the reviewer's corrections. We have changed all the "nor-CPP" into "CPP" throughout the manuscript.

(4) It should be stated more clearly, which compounds are known and which are new. It seems that compound 2 is new, the others are known. For the new compound 2 full analytical data must be given not only NMR data. Please add IR, UV and HRMS data and, most importantly (!), the optical rotation. This is very important, because researchers can correlate to this, in case this compound is isolated again. As the absolute configuration of 2 is known, such a correlation can become highly interesting in the future and will save a lot of unnecessary work, if 2 is isolated again.

Response: We thank the reviewer's careful reading and professional suggestions. We have stated "compound 2 is a new compound, and compounds 1, 3-9 are known" clearly in both introduction and results parts. We totally agree that the complete spectroscopic data of the new compound 2 should be provided. As your request, we have provided the IR, UV, HREIMS, and specific optical rotation data of 2 into the "Methods" part. These added data definitely improved our manuscript towards facilitating future isolation/comparison by other groups.

(5) Line 129: The absolute configuration of the diterpenes 3 – 6 was determined by X-ray. Actually, the observed absolute configurations are already set by the (known) type II diterpene synthase OsCPS4 that was used for the first enzymatic step. The opposite absolute configurations would not be expected. This should at least be discussed. This is only done for compound 2, hidden in the experimental part.

Response: Yes, the absolute configurations of products 2-7 are greatly benefitted by the known absolute configurations of the substrate *syn*-CPP, which is produced by the known class II DTS OsCPS4. We should state this information more clearly in the manuscript. As the reviewer's suggestion, we have added the discussion sentences of "The absolute configuration assignments of 2-7 were also benefitted by the structure of *syn*-CPP, whose absolute configurations have already been set from the catalysis of the class II DTS OsCPS4." in both "Results" and "Methods" parts.

(6) Several mutations of active site residues did not change the productivity of the enzyme, while three changes did. The authors conclude that these residues are involved in catalysis (implying the others are not).

The interpretation should be done here more carefully. What means „involved in catalysis“? Direct interaction (by covalent bonds, hydrogen bonds, non-polar interaction)? For valine a participation in catalysis is difficult to understand. On the other hand, isn't the whole enzyme to be seen as „the catalyst“? In type I terpene cyclisations the enzyme is usually considered to function by ionisation of the substrate, while non-polar active site residues force the substrate in a reactive conformation for the cyclisation cascade that happens more or less independent of the enzyme. Maybe certain residues are relevant for cation stabilisations. The exchange of valine against alanine will likely only decrease the steric bulk in the active site, allowing the substrate to shift a bit within the active site, leading to different deprotonations. A direct involvement in catalysis is not expected.

Response: We thank the reviewer's professional questions, discussion and corrections! We agree that it is not appropriate to conclude the three residues are "involved in catalysis", and we should interpret their functions more carefully and more clearly. The Val84 definitely is not directed involved in the catalysis, and as the reviewer's insightful discussion, it provides steric bulk in the active site to stabilize the CPP-binding for efficient deprotonation and cyclization. We have checked the whole manuscript and adjusted several sentences to make this clear and avoid incorrect descriptions.

(7) Line 344: Reference 10 seems to be wrong here. The authors discuss diterpenoids from plants and fungi, but ref. 10 is from a streptomycete (bacterium).

Response: We thank the reviewer's careful reading and correction. Yes, Ref. 10 is not suitable for citation at this place, and we have replaced Ref. 10 with two literatures related to plant and fungi-originated diterpenoids.

(8) SI Tables S5 and S6: references to previously published NMR data should be given, if available.

Response: We have added all the references related to NMR data of the known compounds in SI Table S5 and S6.

Finally, we would like to take this opportunity to again thank the reviewer for the careful review and excellent guidance about how to improve our study. Many thanks!

REVIEWERS' COMMENTS:

Reviewer #2 (Remarks to the Author):

The authors have addressed all comments satisfyingly and the manuscript can now be accepted.